# Transmission Pathways of the VNN Introduced in Croatian Marine Aquaculture

**DOI:** 10.3390/pathogens11040418

**Published:** 2022-03-30

**Authors:** Snježana Zrnčić, Dragan Brnić, Valentina Panzarin, Miriam Abbadi, Ivana Lojkić, Ivana Giovanna Zupičić, Dražen Oraić

**Affiliations:** 1Department of Pathology, Croatian Veterinary Institute, Savska cesta 143, 10000 Zagreb, Croatia; zupicic@veinst.hr (I.G.Z.); oraic@veinst.hr (D.O.); 2Department of Virology, Croatian Veterinary Institute, Savska cesta 143, 10000 Zagreb, Croatia; brnic@veinst.hr (D.B.); ilojkic@veinst.hr (I.L.); 3Istituto Zooprofilattico Sperimentale delle Venezie, Viale dell’Università 10, 35020 Legnaro, PD, Italy; vpanzarin@izsvenezie.it (V.P.); mabbadi@izsvenezie.it (M.A.)

**Keywords:** VNN, epidemiology, European sea bass, feral fish, surveillance, phylogenetic analysis

## Abstract

Due to the insufficient capacity of Croatian hatcheries, marine aquaculture depends on the importation of fry from different countries in the Mediterranean basin. Importation enables a risk of spreading pathogenic agents. Viral nervous necrosis (VNN), caused by betanodavirus is devastating for the farming of European sea bass. We described a VNN outbreak that occurred in Croatia in 2014. After the diagnosis of VNN in sea bass fry introduced from the same hatchery to five unconnected marine farms at the Adriatic Coast, we performed surveillance within one of the affected farms. It resulted in proven horizontal spreading of the virus within the farm and to feral fish around farm cages. Real-time RT-PCR tested samples showed the dependence of the virus’ proliferation to the water temperature and the fish age. The highest mortality rates were noted during higher sea temperatures. Phylogenetic analysis of partial sequences of RNA1 and RNA2 supported the hypothesis that the virus was introduced to all studied farms from the same hatchery. Moreover, phylogenetic analysis of the whole genome sequences of infected farmed sea bass and thicklip mullet showed high similarity and it is unlikely that infection in Croatian sea bass farms has originated from wild reservoirs, as the first positive record in wild mullet was recorded after the disease outbreak.

## 1. Introduction

Croatian marine aquaculture is a fast-growing economic activity and the quantity of produced European sea bass *Dicentrarchus labrax* (ESB) and gilthead sea bream *Sparus aurata* (GSB) has more than doubled, from productions of 2800 and 2900 tons in 2013 to 6754 and 7780 tons in 2020, respectively [1]. Unfortunately, the capacity of Croatian hatcheries is insufficient to fulfil the needs of the national marine aquaculture. Consequently, most of the quantities required for pen cage production is imported from different countries in the Mediterranean basin such as Italy, Greece or France [2]. The intensive international trade of aquatic animals enables a risk of spreading many known and potential pathogens of aquatic animals [3]. 

One of the serious global viral threats for many different marine and freshwater farmed and wild fish species causing significant economic losses in the aquaculture industry is viral encephalopathy and retinopathy (VER), known also as viral nervous necrosis (VNN) [4]. This neurological disease has been described as particularly devastating for farming of ESB in the Mediterranean basin [5,6,7]. The causative agent of VNN/VER is a small, icosahedral, nonenveloped viral particle (25–30 nm) belonging to the family Nodaviridae, genus *Betanodavirus*. Its genome is composed of two single-stranded positive-sense RNA molecules. The RNA1 segment encodes the RNA-dependent RNA polymerase (RdRp) or protein A while RNA2 encodes the capsid protein (CP) [8]. Additionally, subgenomic RNA called RNA3 is coding for two nonstructural viral proteins (B1 and B2) involved in the regulation of viral replication and suppression of cellular RNA interference [7,9]. Based on the phylogenetic analysis of the T4 variable region within the RNA2 genetic segment it is possible to discriminate four different genotypes: striped jack nervous necrosis virus (SJNNV), tiger puffer nervous necrosis virus (TPNNV), barfin flounder nervous necrosis virus (BFNNV) and red-spotted grouper nervous necrosis virus (RGNNV) [10]. Different genotypes favour different temperature ranges which correlate with distinct geographic distribution and host specificity [11]. Nervous necrosis viruses (NNV) have a segmented structure of their genome and due to the genetic mixing of SJNNV and RGNNV genotypes, reassortant strains are generated and reported [11,12,13,14,15]. Reassortant genotypes have primarily been observed in asymptomatic ESB or causing mild symptoms [15,16,17,18]. In recent years number of VNN outbreaks caused by reassortant strains in GSB has increased [9,14,15,16] causing serious mortalities in younger categories. However, the RGNNV genotype is still the most common genotype described in clinical outbreaks in ESB farming in the Mediterranean region as a cause of significant disease and economic losses [7,11,16,17].

Generally, the disease caused by nervous necrosis virus (NNV) is characterised by changes in skin pigmentation, anorexia, lethargy, hyperinflation of the swim bladder, abnormal swimming behaviour and nervous symptoms due to the lesions in the brain and retina [19]. Most often larvae and juveniles are affected by peracute disease where a sharp increase in mortality is the only noticeable clinical sign [20,21]. In older categories of ESB, mortalities are lower and typical symptoms are loss of appetite, dark pigmentation of the skin, abnormal swimming behaviour such as spiral swimming, blindness and hyper-reactivity after stimuli, traumatic lesions on the jaws, head, eyes and nose, even the whole head due to blindness [22,23,24]. Mortalities could vary from 10 to 100% depending on the age of infected fish and water temperature. 

The virus can be transmitted both vertically and horizontally [4]. Vertical transmission was demonstrated by detection of nervous necrosis virus (NNV) in broodstock’s gonads, ovaries and sperm of different fish species while the horizontal transmission was supported by virus detection in different diseased and asymptomatic farmed and wild fish species, molluscs and gastropods [4,7,16,23,24,25,26].

During the late spring and summer of 2014, several Croatian marine fish farms imported ESB fry from the same hatchery located in the Mediterranean area and, as soon as the sea temperature increased, mortalities and typical clinical symptoms were observed and notified.

The present study aimed to describe a case study of Croatian VNN outbreaks that occurred in 2014 and to demonstrate the horizontal spreading of the virus within infected farms and to wild fish swimming around farm cages.

## 2. Results

### 2.1. Necropsy Showed a Typical Pathology Related to VNN

Results of necropsy in all described case studies observed at farms A, B, C, D and E (Figure 1, Table 1) differed, from lack of any visible changes to the presence of lesions typical for VNN (Figure 2A–D). At farm A, symptoms and increased mortality rate started 6 days after ESB fry transfer from the hatchery and recorded sea temperature of 21 °C. Necropsy revealed only congestion and hyperemia of the brain (Figure 2D). 

In the case of farm B, although the presence of the virus was detected in mid-June without any clinical signs, the onset of typical symptoms (Figure 2A,B,D) and increased mortalities started at the end of July jointly with the growth of sea temperature to 23 °C. 

At the same time, the mortality rate increased drastically in the case of the batch imported into the farm in May 2014 (Figure 3). Mortality rate increased also in other infected generations including ESB fry introduced in the farm in March 2014 and almost year-old ESB introduced in farm B in September 2013. Affected fish displayed loss of appetite, dark pigmentation of the skin, abnormal swimming behaviour such as spiral swimming followed by long periods of lethargy, hyper-reactivity after stimuli, atypical vertical position in the water column and traumatic lesions on the jaws, head, eyes and nose (Figure 2A,B). The symptoms were most pronounced in the batch introduced into farm B in May 2014. Necropsy revealed an empty digestive tract, haemorrhages and hyperemia of the brain. Similar changes were reported in farms C, D and E, and in addition, extensive haemorrhages and necrosis on the head, mouth, opercula, around the mouth, in the mouth and congestion and hyperemia of the brain were also observed. 

During the 1-year surveillance period on farm B, samples collected during the summer months of 2014 showed typical symptoms of VNN, while samples collected during spring 2015 were mostly asymptomatic and those sampled during summer 2015 showed inappetence and spiral swimming. Feral fish did not show any symptoms except the sample of thicklip grey mullet collected in 2015, which showed dark pigmentation of the skin. 

### 2.2. The Results of Virological and Molecular Investigations

All brain samples, from the first cases, collected on farms A, B, C, D and E, caused cytopathic effect (CPE) when inoculated on SSN-1 cell lines except for sample from farm B collected in June (Table 1). At the same time, all pools of brains tested positive for the presence of NNV using the rRT-PCR with Cq values ranging from 11.63 in the case of acute mortalities at farm A, to 31.6 in the case of asymptomatic fish at farm B in June 2014. 

Targeted surveillance undertaken for one year at farm B showed that the virus has not spread to GSB cages; however, it has spread among all ESB cages of different origins and ages (Table 2). Moreover, it was observed that obtained Cq values increased with the decrease of sea temperatures, same as mortality rate as it is visible in the case of fish from the cage M1 (Table 2) and all other infected cages (Figure 3). Samples from the cage M1 were collected and tested using the rRT-PCR in December 2014 with a water temperature of 17.2 °C, and the Cq value was 26.56. In February, with the drop of the sea temperature to 12.5 °C, the Cq value was 34.56 while with the increase of the sea temperature to 14.5 °C in April, the Cq value was 30.1. All molluscs and wild fish samples collected around the farm cages 2 months after the outbreak tested negative. However, a year after the initial infection, among samples of wild fish collected around the farm cages, only the sample of thicklip grey mullet tested positive for NNV with the Cq value of 31.30.

The maximum-likelihood phylogenetic trees based on partial RNA1 and RNA2 segments revealed that Croatian isolates, from affected sea bass and thicklip grey mullet caught in the vicinity of farm B, belonged to the RGNNV genotype and shared high nucleotide similarity (99.88–100% for RNA1; 100% for RNA2). The phylogenies also highlighted that Croatian isolates appeared similar to previously reported isolates from Italian farmed sea bass, namely 318.1.2008, 283.2009 and 214.3.4.2009 (Figure 4A,B) (nucleotide similarity ranging from 99.54–100% for RNA1; 99.8–100% for RNA2). 

The analysis performed on complete RNA1 and RNA2 sequences of strain HRV-105/2014 and HRV-101/2015, isolated from fry ESB and wild fish (thicklip grey mullet) of Farm B, respectively, shared 99.9% nucleotide similarity and further confirmed results obtained from partial sequences (Figure 5A,B). 

Overall, taking into account sequencing data and epidemiological records, our analyses suggest that (i) betanodavirus introduction into Croatian sea bass farms has unlikely originated from wild reservoirs, as the first positive record in wild mullet was after the disease outbreak; (ii) Croatian NNV strains are genetically almost identical to viral strains detected in farmed sea bass in the Mediterranean basin; (iii) the nucleotide identity of Croatian strains highlights the common origin of the infection and hence could support the hypothesis of a possible horizontal spreading of the virus within infected farms. However, it is important to consider that sequences availability is crucial for a reliable reconstruction of the infection chain between farmed and wild animals and from farm to farm. The lack of VNN sequences in public databases strongly limit such inferences; thus, results herein reported have to be cautiously considered. 

## 3. Discussion

Infections with betanodaviruses have a negative impact on the global aquaculture [27] and have also been reported as the cause of losses in farmed ESB either due to mortalities or poor growth rate of fish affected by a chronic course of the disease [28]. Estimated economic burden caused by VNN in small-sized farms increases, even to four times, the total cost of production [29]. All farms included in this study were small-sized farms and they represent important economic activity in rural and outlying island communities. Considering the importance of marine aquaculture in Croatia, we studied an introduction of VNN into Croatian marine farms and the transmission pathways within the farm and its surrounding environment. 

Before the outbreaks described herein, Croatian marine aquaculture has not been affected largely with VNN since 1995 when ESB fry imported into two farms were affected by a new, previously unrecognised nerval disease (unpublished data). Thereafter, there were no notified outbreaks although some authors reported the diagnosis of VNN in farmed and wild fish collected in Croatian waters [9]. 

The results of the present study showed clearly that VNN was introduced into five different, unconnected farms by the importation of infected ESB fry from the same hatchery in the period between May and June 2014. It has been confirmed that vertical transmission of the virus from infected broodstock to offspring due to viral shedding from gonads and gastrointestinal tract is the main mechanism of NNV spreading in several fish species [6,30,31,32,33]. The vertical transmission in the aquaculture industry could be overcome by good biosecurity practices in hatchery-reared larvae and juveniles of some fish species [27]. In the case of the supplying hatchery mentioned in the present study, it seems that biosecurity practices failed and batches of infected fry were distributed to cage farming in different Croatian farms. In support of the fact that all batches of ESB fry were imported from the same hatchery during a relatively short period, as shown in Table 1, phylogenetic analysis performed on partial RNA1 and RNA2 sequences revealed that all six isolates belonged to the RGNNV/RGNNV genotype, with all sequences sharing 100% nucleotide identity for RNA2 and close to 100% identity for RNA1 (Figure 4). That further confirms our statement that one source of infection was responsible for all outbreaks described in the present study.

Nevertheless, clinical symptoms noticeably differed between farm A and farm B. In farm B, fry was imported in May and there were no clinical signs of the infection until mid-July, while in the case of farm A, typical symptoms and mortality occurred several days after seeding fry into cages. The difference might be attributed to the fact that the temperature of the sea at farm A in June was 21 °C, while sea temperature of 18 °C recorded in May in farm B was unfavourable for the virus replication as proved by Toffan et al. [17]. Indeed, in reported experimental trials they assessed the influence of different water temperatures on the mortality and viral load in ESB fry infected with different NNV genotypes and concluded that each genotype behaves differently at water temperatures of 20, 25 and 30 °C. In the case of RGNNV genotype isolates analysed in the present work, all specimens tested positive for NNV using the rRT-PCR; however, the severity of symptoms, viral loads detected in the brain of infected fish and mortality rates increased with the increase of water temperature. A similar feature was observed in the case of farm B where it seemed that NNV was persistent since placing the fish into cages in May until the temperature of the sea started to increase. Real-time RT-PCR analysis performed on fry in June, at a water temperature of 20 °C, tested positive for NNV but there were neither symptoms of VNN nor increased mortalities observed (Table 1). However, with an increase of the water temperature to 23 °C and more during July and August, all typical symptoms of the VNN appeared and mortalities increased sharply (Figure 3). Interdependence of the virus propagation and sea temperature was obvious from the outcomes of the surveillance on farm B. The example of ESB from the cage labelled M1 showed different results when tested by real-time RT-PCR for the detection of NNV from summer to next spring (Table 2). In August, at a water temperature of 23 °C, the Cq value was 12.57; in December at a water temperature of 17.2 °C the Cq value was 26.55; in February at awater temperature of 12.5 °C it was 34.56; and in April at a water temperature of 14.5 °C it was 30.1. Taken together, these results highlighted the fact that symptoms and mortalities vanished with the decrease of the water temperature. However, regarding the detected Ct value of 30 reported in April, it remains to be confirmed whether the increased temperature caused an active replication and therefore the interruption of the latent phase of the virus.

Besides vertical transmission, nodavirus may also infect cultured fish even at the grow-out stages through horizontal transmission [34]. The main factors affecting horizontal transmission are farming conditions, stocking density and temperature [6]. At farm B typical symptoms of the disease and increased mortality rates gradually appeared during the period of higher water temperature in other cages with different ages and origins of ESB (Table 2). It was obvious that fish at younger ages were more severely affected compared to those of older ages, and fry imported to farm B in March 2014 (labelled ESB 17) showed almost the same clinical symptoms and suffered a similar mortality rate as the initially infected group imported from an infected hatchery. Consequently, the result of real-time RT-PCR for the presence of NNV resulted in the Cq value of 15.28. Milder symptoms and lower mortality rates were observed in older ages of ESB, as in the case of 1-year-old fish from cages labelled ESB–A, ESB–OK5 and ESB–OK6 (Table 2 and Figure 3). In these three cages, the results of real-time RT-PCR showed lower viral load expressed as Cq values of 26.7, 34.09 and 33.45, respectively. It should be emphasised that the average body weight of ESB from cage A was 29.39 grams compared to 108.91 and 193.73 grams in the other two cages (OK5 and OK6). These findings support the claim that in earlier developmental stages or younger fish, mortalities are higher, and that in older ages mortalities never reach 100% [22,35]. Interesting is the development of the infection in fry imported from another hatchery into farm B, labelled with asterisk in Table 2 (ESB *). The first batch of the fry was sampled in the transport tank and tested NNV negative. A month later, a sample from the same batch tested NNV positive with a Cq value of 34.92 and it was still infected in the next year, in July 2015, with a Cq value of 28.94. 

During the surveillance in farm B, several batches of Gilthead Seabream (GSB) were also sampled both in 2014 and 2015, and all tested negative for the presence of NNV, contrary to the previously reported finding of the carrier status of this species [24]. Castric et al. (2001) experimentally infected GSB with RGNNV by cohabitation and found out that they could harbour the virus without showing disease symptoms but were capable of infecting others with ESB. Conversely, in our case, none of the GSB analysed tested positive for NNV although they were farmed in an environment with proven horizontal transmission of the virus throughout the farm. It is hard to explain this phenomenon; probably, cages with GSB were apart from the current’s routes or virus loads spread in the environment were not sufficient to infect less susceptible species. It might also be that the prevalence of the virus presence in the GSB population was too low and our sampling efforts were not sufficient [36]. 

Moreover, the phylogenetic analysis, performed on partial and complete sequences of Croatian isolates, highlighted a great nucleotide similarity among strains evidencing their common origin and supporting the hypothesis of a possible horizontal spreading of the virus within infected farms. Furthermore, performed analysis also showed a high similarity of Croatian isolates to other previously reported isolates from farmed ESB; nucleotide similarity ranged from 99.54–100% for RNA1 and 99.8–100% for RNA2. Such a result could indicate that the origin of NNV infection in farmed Croatian ESB probably derived from an introduction of farmed infected fish. However, the lack of sequence availability plays an important role to obtain much more accurate epidemiological information regarding the origin of the betanodavirus infection that occurred in Croatia. Indeed, the only available sequences from farmed fish during the time slot of the outbreaks (2014–2015) were Greek, thus rendering it difficult to deduce other inferences.

During the surveillance at farm B, a collected sample of Mediterranean mussels (*Mytilus galloprovnicialis*) from farm installation tested NNV negative contrary to previously reported findings of several authors from Asia [34], the Mediterranean area [14,37] and even from the Adriatic region [38]. The difference between our research and among mentioned reports is that all of the tested samples were collected either from fish markets or collected in molluscs farms or open sea, but none of them were directly connected to the fish farms. It supports the assumption that NNV is circulating throughout the Mediterranean waters and that it should not be necessarily connected to the infected fish farms, as it seems that the majority of the Mediterranean region is endemic for VNN. However, it is still an open question how clinical outbreaks occur and is it possible that the virus circulating in invertebrates could infect free-living fish and transfer the disease to farmed fish. 

Since the NNV was detected in different feral fish species [9,14,34,39,40] we included in the analysis specimens of wild fish species gathering around the farm cages described in our study. Among sampled feral fish (Table 2), only thicklip grey mullet tested positive for NNV even though bogue was reported in several papers as a possible vector of the virus [9,14,40] and, recently, NNV was also detected in Salema [14] in Greece. There are no reports on the presence of the virus in the annular seabream so far. It is important to highlight that soon after the VNN spread throughout farm B, we collected samples of feral fish and they all tested negative. Nevertheless, in the second year, thicklip grey mullet samples hosted the virus. Bearing in mind the biology of this migratory fish species which resides mostly in coastal areas, we performed a whole genome sequencing aiming at comparing the isolated virus with the one derived from the primarily infected farmed fish. RNA1 and RNA2 sequences of isolates HRV-105/2014 and HRV-101-2015, from ESB and thicklip grey mullet, respectively, were almost identical, suggesting the possible transmission of NNV from farmed fish to wild fish. This is, to the authors’ knowledge, the first molecular evidence of NNV transfer from farmed to feral fish. Until now, all NNV positive fish were sampled separately from the outbreak at the marine fish farm. We could hypothesise that thicklip grey mullet is capable of spreading the infection to naive fish, as it was proved that healthy fish infected in the laboratory conditions with NNV isolated from asymptomatic feral fish developed typical disease symptoms [34]. Additionally, feral fish infected at the fish farms could endanger the wild population and, although the outbreaks in natural populations are scarce, there are several reports on VNN outbreaks and mortalities of endangered species as groupers in the Ionian Sea (Southern Italy) [41]. This finding is supporting a similar hypothesis on the spreading of infection from marine fish farms to the wild population of groupers but the authors had no direct molecular evidence for the transmission since the farm was some 20 miles away from the outbreak position and they had no insight in the health status of the farm. 

## 4. Materials and Methods

### 4.1. Description of Disease Outbreaks

#### 4.1.1. Farm A

At the beginning of June 2014, a clinical outbreak with increased mortalities of European sea bream (ESB) was notified from farm “A” situated near the island Iž in the Middle Eastern Adriatic Sea. Anamnestic data revealed that fry, weighing about 8.0 grams, were imported from an abroad hatchery a few days ago. The sea temperature was 21 °C and affected fish were reluctant to feed, exhibited abnormal swimming behaviour and died with gradually increasing mortalities (Figure 1, Table 1). A farmer-collected sample consisting of 30 specimens of ESB fry from the cage where clinical signs were noticed. They tried to sample mostly individuals with changes in behaviour, packed them in the cooling transport box and delivered them to the laboratory under cooling conditions.

#### 4.1.2. Farm B

In June 2014, farm B, situated about 48 km south-east (Figure 1) from farm “A”, sent ESB fry samples (weighing about 12.5 grams) for regular health controls. ESB fry originated from the same abroad hatchery that supplied farm A. There were neither changes in behaviour, nor increased mortalities. The sea temperature was 21 °C. Since there were neither signs of abnormal behaviour nor increased mortality, the farm manager randomly collected 30 individuals of fry imported to the farm in May from the hatchery XY. A month later, at the end of July and beginning of August, the same ESB fry batch (weighing now about 19.5 grams) manifested increased mortality and changes in behaviour and appearance (Figure 2 and Figure 3, Appendix A). A sample consisting of 30 individuals of affected fish was collected (Table 1). At the same time, they collected a sample consisting of 30 individuals of fry imported in September 2013 from the hatchery labelled FZ (Table 2). Both samples were packed in the cooling transport box and sent for diagnostics under cooling conditions. 

#### 4.1.3. Farm C

At the beginning of August, continuously increased mortalities of ESB fry (weighing about 17.5 grams) imported from an abroad hatchery XY were noticed and the cause was attributed to a visible heavy infection with *Diplectanum aequans* (Appendix A). The temperature of the sea was 24 °C. Farmers tried to cure parasitic disease using formalin bath and as increased mortalities persisted, they collected a sample consisting of 30 individuals showing disease symptoms, packed them in the cooling box and sent them for diagnostics under cooling conditions (Table 1, Figure 1).

#### 4.1.4. Farm D and Farm E 

Farms D and E, both small family-owned farms, imported ESB fry from the same abroad hatchery XY in the second part of May 2014. Farm D is situated close to island Pašman in the middle Eastern Adriatic, while farm E is situated further south, in the estuary of the river Krka, in the area of brackish water (Figure 1). Starting from the beginning of August, both farms suffered increased mortalities. In farm D, infection with isopod parasite, *Ceratothoe oestroides* was observed and mortalities were attributed to the parasite infection, while mortalities in farm E were all the time unexplained. Repeated unsuccessful treatments led farmers to send samples for diagnostics belatedly, after suffering high losses of ESB fry. Each of them collected 30 individuals showing typical disease symptoms, packed them in the cooling boxes and delivered them for laboratory analysis under cooling conditions.

#### 4.1.5. Targeted Surveillance at Farm B

After NNV detection in farm B, 1-year targeted surveillance was performed. Farm B was selected for this study because of their willingness to share all production data with the diagnostic laboratory openly, precise recording of the environmental conditions and capacity to collect samples of wild fish in the vicinity of the farm. A mutual plan of sampling and notification of the changes and environmental conditions was set up with the farm manager. Sea temperatures were recorded, aiming to find out the pathways of the virus spreading among ESB and gilthead sea bream (GSB) cages on the farm. Besides surveillance, noting of daily mortality in each cage is a routine procedure and data were used in this study (Figure 3 and Appendix A). Farm B is a small family farm supplying local markets and restaurants. To be able to supply a market size fish all over the year, ESB and GSB are usually purchased from two different hatcheries, herein labelled as XL and FZ (Table 2), and are seeded in spring and autumn. During the surveillance period both ESB and GSB, of different ages and different cages, were sampled. Additionally, newly imported fry was sampled from transportation tanks before being seeded into farm cages. Each sample collected during the surveillance period consisted of 30 fish, packed in the cooling boxes and delivered to the laboratory under cooling conditions. During the surveillance, pools of 10 brains were subjected to real-time RT-PCR for NNV detection. Both in 2014 and 2015, samples of mussels collected from the cages and feral fish caught around the cages, namely bogue *Boops boops* (*n* = 5 in 2014 and *n* = 4 in 2015), thicklip grey mullet *Chelon labrosus* (*n* = 4 in 2014 and *n* = 4 in 2015), salema *Sarpa salpa* (*n* = 4 in 2014 and *n* = 3 in 2015) and annular seabream *Diplodus annularis* (*n* = 3 in 2015) were included in the NNV targeted surveillance. 

### 4.2. Necropsy 

The affected fish from all described outbreaks were transported to laboratories, at refrigeration temperature (from 3 to 8 °C), for further diagnostic investigations. 

Upon receipt of samples, fresh preparations of skin, gills, intestines, kidneys and swim bladders in the form of smears, scrapes, and/or squashes were examined microscopically as unstained preparations for the presence of parasites or other recognizable abnormal features. Internal organs were examined for the presence of gross lesions, and then swabs were taken from the head, kidney, spleen and heart for bacteriological examination. In all outbreaks, samples of the brain were collected for virological examination and molecular analysis. Tissue homogenates for molecular analysis were tested immediately. Only during the first suspicion of VNN on each farm were samples subjected to the virological investigation, while during the surveillance, only molecular analyses were applied.

### 4.3. Virological Investigations

On the basis of the clinical signs and necropsy findings, nervous necrosis virus infection was suspected. Brain samples were collected and pooled from five specimens, homogenised using a mortar and pestle with sterile sand and suspended 1:10 in Leibowitz 15 (L15, Lonza, Basel, Switzerland) supplemented with 10% *v*/*v* foetal calf serum (FCS, BI, Kibbutz Beit-Haemek, Israel) and 2% *v*/*v* antibiotic-antimycotic solution (Penicillin 100 UI/mL, Streptomycin sulphate 10 mg/mL, Amphotericin B 25 μg/mL and Kanamycin 10 mg/mL) (Merck, Darmstadt, Germany). Homogenised samples were centrifuged at 3634× *g* for 30 min at 4 °C and tissue extracts were incubated overnight in refrigerator at 4 ± 2 °C. Stripped snakehead cell lines (SSN-1) [26] were grown in 25 cm² tissue culture flasks (Euro Clone, Milan, Italy) in an L15 cell culture medium supplemented with 10% FCS, 1% L-glutamine, and 1% antibiotic-antimycotic solution without Kanamycin (Sigma Aldrich, St. Louis, MI, USA) at 25 °C. Tissue extracts were inoculated at two 10-fold dilutions (1:10 and 1:100) onto 1-day-old SSN-1 cells grown in 24-well cell culture plates (Thermo Scientific Nunc, Waltham, MA, USA) and incubated at 25 °C. After inoculation, plates were observed daily for detection of a cytopathic effect (CPE). After 7 days, supernatants were filtered through 0.45 μm membranes and used to inoculate actively growing SSN-1 cells. Samples were examined along another progressive passage of 7 days.

### 4.4. Real-Time RT-PCR and Partial Amplification of the RNA1 and RNA2 Genomic Segments

Tissue homogenates prepared for virological investigation were also tested using a real-time RT-PCR (rRT-PCR) [42] targeting betanodavirus RNA2 segment encoding for capsid protein. Briefly, RNA was extracted from 400 µL tissue supernatant using iPrep™ PureLink™ Virus Kit (ThermoFisher Scientific, Waltham, MA, USA), according to the manufacturer’s instructions. After RNA purification, real-time RT-PCR was conducted on RotorGene Q (Qiagen, Hilden, Germany) by using QuantiFast Pathogen RT-PCR + IC Kit (Qiagen, Germany) following the manufacturer’s recommendations regarding reaction composition and cycling conditions.

Selected NNV positive samples representative of each farm and period were used for the partial RNA1 and RNA2 segment amplification by conventional RT-PCR [11]. For this purpose, the OneStep RT-PCR kit (Qiagen, Hilden, Germany) was used according to manufacturer instructions with annealing temperature described by Panzarin et al. [9] and RT-PCR reactions performed on ABI 9700 GeneAmp thermal cycler (Applied Biosystems, Waltham, MA, USA).

### 4.5. Whole Genome Sequencing

Two NNV positive samples from Farm B, originating from ESB fry and grey mullet, were submitted to whole genome sequencing. Since these samples were confirmed to be RGNNV/RGNNV genotype from the previous step of partial RNA1/RNA2 segment amplification, a specific set of primers (nine primer pairs) was used to cover the complete sequence of both genomic segments (procedure available on request). The amplification was performed on ABI 9700 GeneAmp thermal cycler (Applied Biosystems, Waltham MA, USA) using SuperScript™ III One-Step RT-PCR System with Platinum™ Taq DNA Polymerase (ThermoFisher Scientific, Waltham, MA, USA). Thermal cycling conditions for all RT-PCR reactions were as follows: reverse transcription at 55 °C for 30 min, initial denaturation at 94 °C for 2 min, 40 cycles of denaturation at 94 °C for 1 min, annealing at 50 °C for 1 min, elongation at 68 °C for 1 min; the reactions were terminated with 5 min elongation at 68 °C.

### 4.6. Sequencing and Phylogenetic Analysis

RT-PCR products were purified before sequencing using the ExoSAP-IT™ PCR Product Cleanup Reagent (ThermoFisher Scientific, Waltham, MA, USA) under the following modified conditions: 2 µL of ExoSAP-IT reagent was added to 20 µL of RT-PCR product and firstly incubated at 37 °C for 45 min and further at 80 °C for 15 min. Direct Sanger sequencing was performed by Macrogen Europe (Amsterdam, The Netherlands). Sequences of the partial RNA1 and RNA2 segments were deposited at NCBI GenBank under accession numbers OM513976-OM513980 and OM513983-OM513987, respectively. Complete CDS of two strains were deposited under accession numbers OM513981-OM513982 and OM513988-OM513989 for RNA1 and RNA2 segments, respectively.

RNA1 and RNA2 phylogenetic analysis were performed for both partial consensus of ESB samples HRV-67/2014, HRV-105/2014, HRV-109/2014, HRV-111/2014, HRV-130/2014, HRV-141/2014 and grey mullet HRV-101/2015, and complete sequences obtained for HRV-105/2014 and HRV-101/2015.

Consensus sequences were aligned and compared to a selection of reference sequences retrieved from GenBank using MEGA 7.0 software [43]. The selection of reference sequences for the dataset construction was performed according to (i) the BLAST results obtained for each gene of each sample and (ii) a selection of isolates derived from Mediterranean wild and farmed fish from 2008 to the year of the outbreaks (2014). For both segments, phylogenetic relationships among the isolates were inferred using the maximum-likelihood (ML) method available in the IQ-Tree v1.6.9 program [44]. The best-fitting model of nucleotide substitution was determined with Model Finder [45]. One thousand bootstrap replicates were performed to assess the robustness of individual nodes of the phylogeny and only values ≥ 70% were considered significant. Phylogenetic trees were visualised using the FigTree v1.4.2 program (http://tree.bio.ed.ac.uk/software/figtree/) (accessed on 19 February 2022).

## 5. Conclusions

The results of the surveillance on the fish farm described in this study are fully in line with the results of studies on the epidemiology of VNN carried out in laboratory conditions. Once NNV is introduced into the marine fish farm, the virus spreads very quickly among susceptible hosts and horizontal transmission was confirmed. Transition rates of NNV between different farming units are corelated to environmental factors, mainly water temperature, an inevitable trigger of the virulent expression, clinical disease development and high mortality rate. Although we demonstrated transmission of the NNV from farmed ESB to wild fish using whole genome sequences of the isolates, in future research a real risk of infection spreading from wild to farmed fish should be evaluated. Unfortunately, currently, the insufficient number of available sequences from the Mediterranean area in public databases limits reliable reconstruction of the infection chain between farmed and wild animals and from farm to farm.

## Figures and Tables

**Figure 1 pathogens-11-00418-f001:**
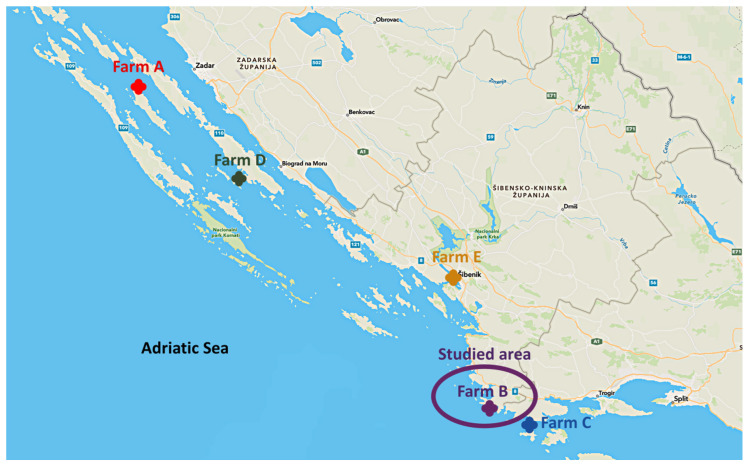
Map of Middle Eastern Adriatic Sea with farms included in the study.

**Figure 2 pathogens-11-00418-f002:**
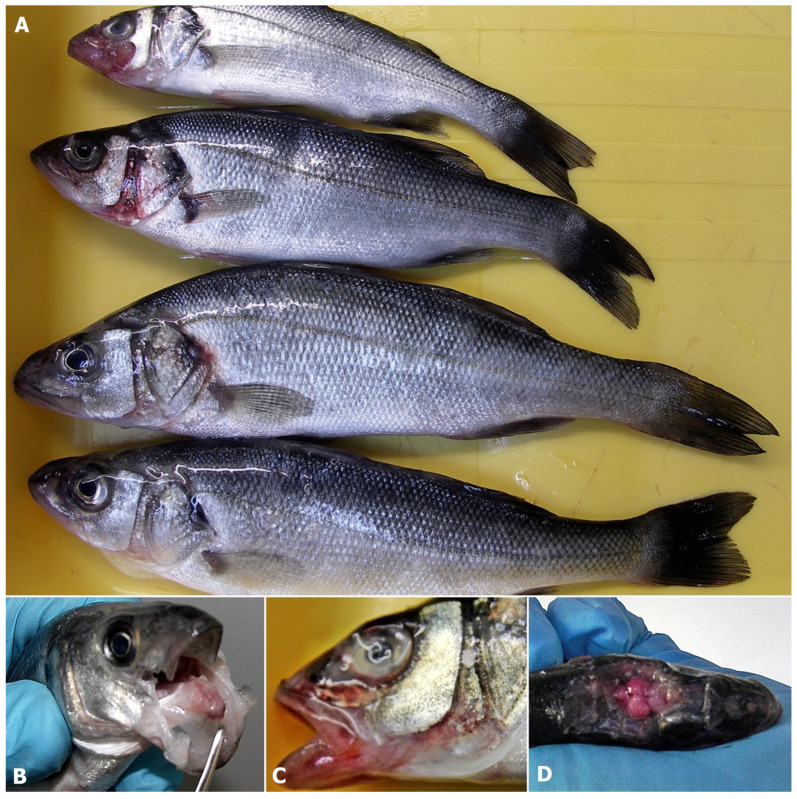
Typical clinical symptoms observed in ESB infected with NNV presented as dark pigmentation of the skin, traumatic lesions on the jaws, nose and eyes (**A**), erosion and haemorrhages in the mouth (**B**), erosion, scale loss on the head and corneal opacity (**C**), congestion and hyperemia of the brain (**D**).

**Figure 3 pathogens-11-00418-f003:**
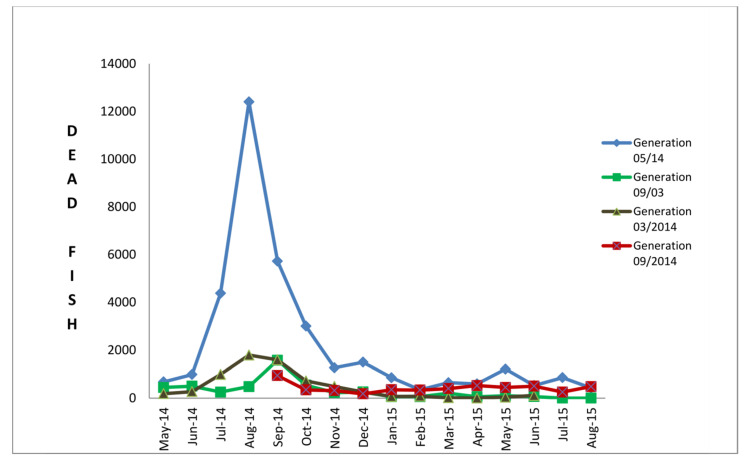
The cumulative mortality rate in four different generations of ESB within farm B over surveillance period (June 2014 –August 2015). Four different generations of fry were surveilled for mortalities, namely, generation seeded in May 2014 (initially infected), September 2013, March 2014 and September 2014.

**Figure 4 pathogens-11-00418-f004:**
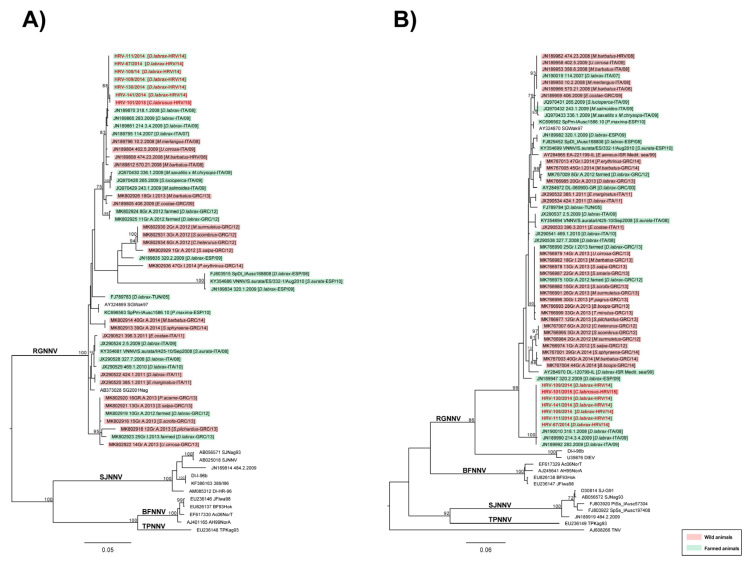
ML phylogenetic trees of Croatian *Betanodavirus* strains based on partial RNA1 (**A**) and RNA2 (**B**) sequences. The phylogenies included betanodaviruses derived from Mediterranean wild and farmed fish from 2008 to 2014. For each strain information regarding species, country and year of isolation are reported within brackets. Croatian sea bream and thicklip grey mullet betanodaviruses under investigation are highlighted in red. Strains isolated from wild fish are labelled in pink, while betanodaviruses detected in farmed fish are labelled in green. The genotype subdivision according to Nishizawa et al. (1997) is shown at the main branches. The numbers at the nodes represent bootstrap values (only values ≥ 70% are reported), while branch lengths are scaled according to the number of nucleotide substitutions per site. The trees are midpoint rooted for clarity only. The scale bars are reported.

**Figure 5 pathogens-11-00418-f005:**
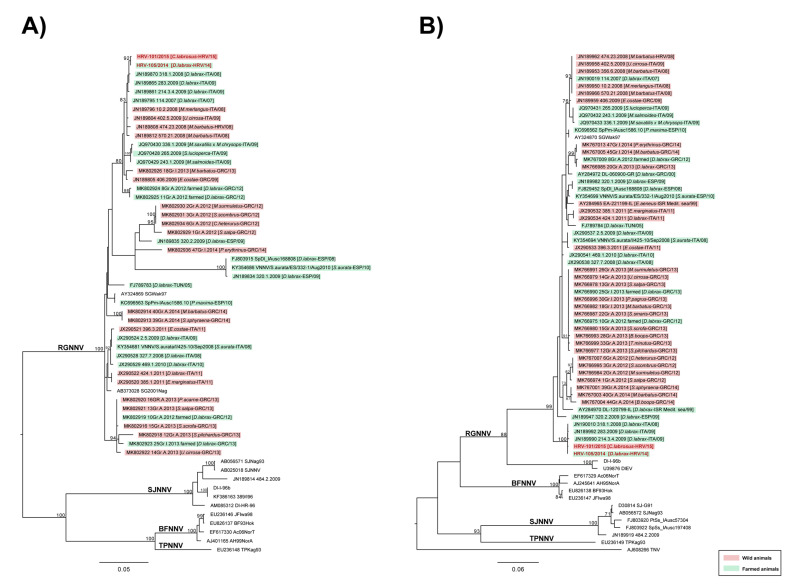
ML phylogenetic trees of Croatian *Betanodavirus* strains based on complete RNA1 (**A**) and RNA2 (**B**) sequences. The phylogenies included betanodaviruses derived from Mediterranean wild and farmed fish from 2008 to 2014. For each strain information regarding species, country and year of isolation are reported within brackets. Croatian sea bass and thicklip grey mullet betanodaviruses under investigation are highlighted in red. Strains isolated from wild fish are labelled in pink, while betanodaviruses detected in farmed fish are labelled in green. The genotype subdivision according to Nishizawa et al. (1997) is shown at the main branches. The numbers at the nodes represent bootstrap values (only values ≥ 70% are reported), while branch lengths are scaled according to the number of nucleotide substitutions per site. The trees are midpoint rooted for clarity only. The scale bars are reported.

**Table 1 pathogens-11-00418-t001:** Farms with clinical cases of VNN occurred during 2014.

Farm	Date of Fry Import	Sea Temperature °C	Sampling Date	ANALYSIS CODE	Fish Weight (g)	Sea Temperature °C	rRTPCR (Cq)	SSN1 Cell Culture
A	07/06/2014	20	13/06/2014	67/14	8.0	21	11.63	CPE*
B	13/05/2014	18.5	23/06/2014	71/14	12.5	21	31.6	No CPE
B	13/05/2014	18	03/08/2014	105/14	19.5	23	12.57	CPE
C	18/05/2014	16	04/09/2014	111/14	17.5	24	19.2	CPE
D	06/06/2014	20	10/10/2014	130/14	50.8	22	17.57	CPE
E	06/06/2014	20	24/10/2014	141/14	84.3	21	19.8	CPE

CPE*—cytopathic effect.

**Table 2 pathogens-11-00418-t002:** Results of NNV spreading within farm B over 1-year period surveillance (June 2014 –August 2015).

Sampling Date	Fish Species—Cage Label	Origin	Date of Import	Analysis Code	Fish Weight (g)	Fish Age	Sea Temperature °C	rRT-PCR (Cq)
23/06/2014	ESB–C	Hatchery XL	05/2014	71/14	12.5	Fry	21	31.6
03/08/2014	ESB–M1	Hatchery XL	05/2014	105/14	19.5	Fry	23	12.57
03/08/2014	ESB–A	Hatchery FZ	09/2013	105/14	29.39	Year	23	26.71
18/08/2014	ESB–17	Hatchery FZ	03/2014	109/14	44.6	Fry	25	15.28
18/08/2014	ESB–OK6	Hatchery FZ	09/2013	109/14	108.91	Year	25	34.09
18/08/2014	ESB–OK5	Hatchery XL	05/2013	109/14	193.73	Year	25	33.45
18/08/2014	GSB–OK3	Hatchery FZ	03/2014	109/14	47.42	Fry	25	Neg.
18/08/2014	GSB–14	Hatchery FZ	04/2014	109/14	38.36	Fry	25	Neg.
18/08/2014	GSB–15	Hatchery FZ	04/2014	109/14	39.57	Fry	25	Neg.
05/09/2014	ESB *	Hatchery FZ	09/2014	115/14	7.8	Fry	24	Neg.
05/09/2014	Bogue (*Boops boops*)	Caught close to the farm	n/a	115/14	139	n/a	24	Neg.
05/09/2014	Thicklip grey mullet (*Chelon labrosus*)	Caught close to the farm	n/a	115/14	447	n/a	24	Neg.
05/09/2014	Salema (*Sarpa salpa*)	Caught close to the farm	n/a	115/14	210	n/a	24	Neg.
05/09/2014	Mussel (*Mytilus galloprovincialis)*	Collected from the cages	n/a	114/14	n/a	n/a	24	Neg.
22/09/2014	ESB–C	Hatchery FZ	09/2014	125/14	8.3	Fry	23	37.22
03/12/2014	ESB–M1	Hatchery XL	05/2014	162/14	73.29	Fry	17.2	26.56
03/12/2014	ESB–B	Hatchery XL	05/2014	162/14	72.43	Fry	17.2	25.73
25/02/2015	ESB–M1	Hatchery XL	05/2014	12/15	85	Year	12.5	34.56
25/02/2015	ESB–M2	Hatchery XL	05/2014	12/15	74.31	Year	12.5	33.27
02/04/2015	ESB–M1	Hatchery XL	05/2014	23/15	61.7	Year	14.5	30.1
30/04/2015	ESB *	Hatchery FZ	04/2015	36/15	3.28	Fry	16	Neg.
27/05/2015	ESB–C1	Hatchery FZ	04/2015	56/15	5.86	Fry	19	Neg.
27/05/2015	ESB–C2	Hatchery FZ	04/2015	56/15	4.9	Fry	19	Neg.
17/06/2015	ESB–C1	Hatchery FZ	04/2015	67/15	8.53	Fry	23	35.85
17/06/2015	ESB–C2	Hatchery FZ	04/2015	67/15	7.96	Fry	23	34.92
02/07/2015	ESB–C1+C2	Hatchery FZ	04/2015	72/15	9.04	Fry	23	Neg.
02/07/2015	ESB–OK1	Hatchery FZ	09/2014	72/15	81.23	Year	23	28.94
02/07/2015	ESB–OK4	Hatchery XL	05/2014	72/15	159.5	Year	23	26.67
12/08/2015	ESB–OK1	Hatchery FZ	09/2014	101/15	151	Year	26	Neg.
12/08/2015	ESB–OK6	Hatchery FZ	04/2014	101/15	240.5	Year	26	30.82
12/08/2015	ESB–OK2	Hatchery XL	05/2014	101/15	192.5	Year	26	33.44
12/08/2015	ESB–OK4	Hatchery XL	05/2014	101/15	189	Year	26	31.25
12/08/2015	Bogue (*Boops boops*)	Caught close to the farm	n/a	101/15	248	n/a	26	Neg.
12/08/2015	Salema (*Sarpa salpa*)	Caught close to the farm	n/a	101/15	485	n/a	24	Neg.
12/08/2015	Thicklip grey mullet (*Chelon labrosus*)	Caught close to the farm	n/a	101/15	280	n/a	24	31.30
12/08/2015	Annular seabream (*Diplodus annularis*)	Caught close to the farm	n/a	101/15	97	n/a	24	Neg.

*: Samples of fry collected from transport tank.

## Data Availability

The data presented in this study are available upon a reasonable request from the corresponding author.

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
