# Peer review of "Transmission Pathways of the VNN Introduced in Croatian Marine Aquaculture"

_pathogens, 2022, doi:10.3390/pathogens11040418_

Round 1

Reviewer 1 Report

The present manuscript on Transmission pathways of the VNN introduced in Croatian marine aquaculture is well described and timely required for the researchers working on fish diseases and an important viral disease outbreak in mediterranean farms. The manuscript is well described and should be considered after minor revisions. 

The sampling conditions from different farms are not well described and must be improved. 

It is important to highlight why farm B was targeted for surveillance?

The quality of figure 1 and 4 should be improved.

Reviewer 2 Report

In this manuscript, viral nervous necrosis (VNN) outbreak that occurred on five farms in Croatia in 2014 was described in detail through epidemiological surveys, virological testing and molecular evolutionary analysis. The survey results provide some reference for the healthy development of aquaculture in Croatia.

Questions and suggestions:

1、Why farm B was chosen for surveillance instead of other farms?  Symptoms of diseased fish from farm A were significantly earlier and more typical than those from farm B.

2、Chinese characters should not appear in Figure 3 (eg May 14) and the generations represented by gray and yellow lines should be supplemented.  

3、CPE observation of SSN-1 cells after inoculation with diseased fish brain homogenate should be provided.

4、In this investigation, what is the physical condition of broodstock in the hatchery? Do they present with clinical symptoms or test positive for NNV?

5、What has been the prevalence of VNN per farm in recent years?
